# Social Inequalities in Dog Bites and Strikes in Scotland: Evidence from Administrative Health Records and Implications for Prevention Policy

**DOI:** 10.3390/ani15131971

**Published:** 2025-07-04

**Authors:** Jade Hooper, Hannah M. Buchanan-Smith, Tony Robertson, Paul Lambert

**Affiliations:** 1Faculty of Social Science, University of Stirling, Stirling FK9 4LA, UK; paul.lambert@stir.ac.uk; 2Faculty of Natural Sciences, University of Stirling, Stirling FK9 4LA, UK; h.m.buchanan-smith@stir.ac.uk (H.M.B.-S.); tony.robertson@stir.ac.uk (T.R.)

**Keywords:** dog bites, wounds and injuries, socioeconomic factors, health inequalities, social determinants of health, public health, Scotland, emergency medical services, hospitalisation, administrative health data

## Abstract

In the UK, there is growing concern about injuries from dog bites and strikes (DBS) and how best to prevent them. In this paper, we examine administrative health records in Scotland about incidents when an injury was caused by a dog. The analysis of that data allows us to assess trends in the number of incidents and selected social circumstances associated with them. We find that, in Scotland between 2007 and 2019, there was some growth in the rate of injuries, although some of this reflects improved reporting, rather than more incidents. More importantly, we find that there are strong social patterns in DBS incidents, for instance, that younger people are more often involved and that incidents more often occur in areas with higher deprivation. In light of such patterns, we discuss how policies to prevent dog-related injuries would benefit from thinking more holistically about social inequalities in risks, in a similar way to how other social inequalities in public health are recognised and responded to.

## 1. Introduction

In the UK, hospital admissions due to dog bites and strikes (DBS) are rising [1,2,3]. Injuries may be severe and long-lasting, often leading to significant emotional distress and psychological consequences such as post-traumatic stress disorder, phobias, nightmares, and anxiety [4]. Dog bite incidents, therefore, represent a significant physical and mental health concern for victims and their families, an issue described as a ‘national crisis’ in Scotland [5].

It is well documented that adverse health outcomes are unevenly distributed, disproportionately affecting the most socially disadvantaged communities [6]. A prominent area of health inequality research concerns ‘unintentional injuries’ (UIs)—for instance, road traffic injuries, burns, falls, drowning, and poisoning [7]. UIs are a leading cause of death and disability [8], for children in particular [9]. Once referred to as accidents, these injuries are now understood to be largely predictable and preventable [7], shaped by broader social and environmental factors [10].

Dog bites constitute a type of preventable and predictable UI [11,12]. Whilst the World Health Organisation (WHO) has highlighted dog bites as a significant public health problem, with an unfair burden on children in low- and middle-income countries [13], dog bites are rarely considered within mainstream UI or health inequalities research, and few studies examine DBS prevention through an equity lens. Nonetheless, social inequities in dog bite injuries have already been documented in the UK [1,3,14,15,16] and internationally [17,18,19,20,21,22,23,24,25,26,27,28]. Moreover, causal pathways related to inequality may be relevant to strategic thinking surrounding DBS prevention (such as limited resources, compromised parental supervision, and more hazardous home or neighbourhood environments [29]).

In spite of increasing recognition of the underlying structural and social determinants of dog bite injuries [30], UK policy efforts to address dog bites have largely ignored inequalities in favour of punitive legislative approaches. The Dangerous Dogs Act (1991) restricts the ownership of certain breeds. The Control of Dogs (Scotland) Act 2010 and the Anti-Social Behaviour, Crime and Policing Act 2014 (England and Wales), and recent changes to the Dangerous Dogs Act through the addition of the XL Bully breed in England and Wales (The Dangerous Dogs (Designated Types) (England and Wales) Order 2023) and Scotland (The Dangerous Dogs (Designated Types) (Scotland) Order 2024) have re-ignited public and academic debate but have rarely acknowledged the wider social context or inequities that shape risk exposure. Breed-specific legislation in general has been criticised for being a rushed response to high-profile incidents, lacking a robust evidence base, and failing to address the complex, multifactorial nature of dog bite risk (e.g., [31]). Scholars increasingly argue that there is a need for less punitive and more preventative approaches to policy [17,32], advocating for a public health approach that considers not only individual-level behaviour but also the broader determinants of health [30]. However, this approach has yet to be implemented in practice.

Some researchers have, though, begun to apply social determinants frameworks to dog bite risk. For instance, Watkins and Westgarth [30] adapted the Dahlgren and Whitehead [33] ‘rainbow model’ to illustrate the multiple levels of influence on dog bite risk. Others have drawn on child injury prevention models to conceptualise at-risk interactions between children and dogs [34]. Duncan-Sutherland et al. [11] also emphasise the importance of macro-level research, while Hersant et al. [26] advocate for the adoption of lessons from other injury prevention campaigns.

Despite this emerging evidence base, empirical studies explicitly linking dog bite risk to social inequalities remain limited. Several UK studies have documented a social gradient in hospital admissions for DBS, with admission rates two to three times higher in the most deprived areas compared to the least deprived [1,3,14,15,16]. However, these studies have focused almost solely on hospital admissions, which tend to capture more severe injuries and may underestimate the broader burden of dog bites. Such limitations in evidence hinder the development of informed, equitable prevention strategies, and risks perpetuating existing disparities [7].

A key challenge in this research area is the inconsistent definition and documentation of dog bite incidents. The definition of a dog bite or dog bite incident is not always well defined, and views on what does or does not constitute a dog bite often vary [35]. In Scotland, as is the case for many countries, dog bites are not routinely documented in any reliable or purposeful way. As such, it is typical for research to use administrative health records from injuries caused by dogs to estimate the extent of dog bites in a population. In these cases, a dog bite is defined largely through clinical codes denoting the cause of an injury as a ‘dog bite or strike’ (DBS). Health records containing these codes are then used to explore some of the characteristics of individuals involved. Although not formally defined, a dog ‘strike’ is likely a catch-all for other non-bite dog-related injuries, e.g., being pulled or knocked over or scratched by a dog [35]. The current study used health records involving DBS to identify incidents across Scotland.

Focusing on Scotland remains particularly important due to its unique policy landscape (including The Control of Dogs (Scotland) Act 2010) and recent changes to policy across the UK. Additionally, Scotland has among the most pronounced health inequalities in the UK and in Western Europe [36], making it a relevant context for investigating how dog bite risks intersect with socioeconomic deprivation. Accordingly, this study builds on emerging evidence by examining dog bite-related administrative health records in Scotland across multiple levels of healthcare engagement, thereby capturing a wide range of injury severity and help-seeking behaviours.

## 2. Materials and Methods

### 2.1. Data Access and Ethical Approval

Health records were obtained with the permission of the National Health Service (NHS) Scotland Public Benefit and Privacy Panel for Health and Social Care. Ethical approval was obtained from the University of Stirling General University Ethics Panel (reference number GUEP367). Data access was facilitated by the Electronic Data Research and Innovation Service (eDRIS).

Due to the sensitive nature of the data, access was only permitted within a secure environment known as a data-safe setting, which was accessed remotely due to COVID-19 restrictions. For this project, the Scottish National Safe Haven was used. Managed by eDRIS and overseen by the Scottish Government Statistics Public Benefit and Privacy Panel, the National Safe Haven is operated by the University of Edinburgh.

All analysis outputs have been subjected to stringent statistical disclosure control checks designed to prevent the identification of individuals and the inadvertent release of personal information.

### 2.2. Health Data Sources and DBS Records

Three sources of health data were explored: NHS 24 (telephone) calls; accident and emergency (A&E) attendances; and Scottish Mortality Records’ general acute inpatient and day cases (SMR01). NHS 24 and A&E records covered the period of 1 January 2011–31 August 2019. SMR01 records covered the period of 1 January 2007–31 August 2019.

Relevant health records were identified using different criteria for each source, depending on the information available within the health datasets. The clinical code W54 ‘bitten or struck by dog’ from The International Statistical Classification of Diseases and Related Health Problems 10th Revision (ICD-10) coding system [37] was used where available. Information from free text variables was also used where possible.

#### 2.2.1. NHS 24 Calls

NHS 24 call records capture initial queries to UK health services via self-referral at any level of severity. These records do not contain ICD-10 codes, and instead, the identification of DBS injuries relied entirely on free text variables detailing the call reason (‘NHS 24 Call Reason’) or the symptoms based on the protocol algorithm launched during the call (‘NHS 24 Symptoms’/‘Protocol Title’). The exact search term ‘DOG BITE’ was used for the symptoms/protocol title field, while the character string ‘%DOG%’, was searched for within the call reason variable. The latter of these resulted in some records which contained the character string for dog but were not related to dog bites and strikes once the free text field was examined (e.g., the human ingestion of dog medication). These records were later filtered out of the analytical dataset.

#### 2.2.2. A&E Attendances

Accident and emergency attendances in the UK can arise for incidents in which hospital treatment is sought, by medical staff or via self-referral, and for mild or severe injuries. To identify A&E attendances for DBS, a combination of variables containing ICD-10 codes and free text information was used. To identify records using ICD-10 codes, the variable ‘Diagnosis on Discharge Code 1’ was searched for codes W54.0-9. Free text variables (‘Diagnosis Group Text 1’ and ‘Presenting Complaint’) were also used to identify additional records within the A&E attendances using the search terms ‘%DOG BITE%’, ‘%Dog bite%’, and ‘%dog bite%’. The majority (81.6%) of A&E attendance records for DBS were identified using the free text variables, rather than through ICD-10 codes.

#### 2.2.3. SMR01 Hospital Admissions

Hospital admissions generally occur only for more severe issues about which medical staff diagnose that sustained support is required. Records for hospital admissions involving DBS were identified solely through ICD-10 codes (W54.0-9), as no additional free text information was available. The variables used were the mandatory field ‘Main Condition’ and the optional fields ‘Other Condition (1–6)’. None of the records contained ICD-10 codes for DBS in the main condition field but were all extracted using information from the optional secondary fields.

### 2.3. Other Measures

The year of record and age of respondents were recorded in all datasets. Information on the location of the incident was available for some A&E attendance records via a specific variable (‘place of occurrence’), although this field was frequently missing or non-specific. Where possible, location was supplemented or derived from the fourth character of ICD-10 codes in both the A&E attendance and SMR01 hospital admission records. However, a large proportion of records in both datasets still had missing, unspecified, or vague location information.

The Scottish Index of Multiple Deprivation (SIMD) is a relative measure of area-based deprivation using data on seven domains: income; employment; health; education, skills and training; geographic access to services; crime; and housing. The SIMD is measured at the small-area level (‘datazones’), ranked from most to least deprived, and often divided into deciles or quintiles across Scotland. SIMD measures designed at different time periods were provided with the health datasets. In analysis, SIMD 2009 was used for health records dated from 2007 to 2009, SIMD 2012 was used for health records dated between 2010 and 2013, and any records occurring in 2014 or later used SIMD 2016. SIMD 2016 was used to explore the individual SIMD domains.

Population-based rates per 10,000 persons were calculated using mid-year population estimates time series data (1981–2021) for the year of health record, age, and local authority [38]. Population-based rates via SIMD were calculated using National Records Scotland (NRS) population estimates for SIMD 2020v2 deciles [39], as equivalent estimates for SIMD 2016 deciles were no longer readily available.

### 2.4. Analytical Methods

The health data were provided at the record level, meaning that individuals could have multiple records within each of the datasets. To describe the characteristics of records involving DBS, descriptive outputs are shown on the number and/or rate of records by year of health record, age, SIMD decile, SIMD decile stratified by age bands, SIMD domain, local authority, and local share of the most and least deprived areas. When relevant, 95% confidence intervals for rates are calculated based on the standard errors of a Poisson model predicting the number of incidents. Due to partial data for 2019 (up to September 2019), some outputs exclude this year, which is indicated where relevant.

Data cleaning, the preparation and analysis of micro-level data, was carried out within the National Safe Haven using StataMP 16. Aggregate counts not at risk of statistical disclosure were removed from the Safe Haven when appropriate. Data visualisation of aggregate information was completed with the National Safe Haven using RStudio 2023.12.1 Build 402 “Ocean Storm”. ChatGPT (GPT-4, OpenAI) was used to assist in writing code to generate data visualisations. Local authority boundary files [40] were used to visualise the local authority rates of DBS. Geographic Information System (GIS) mapping was carried out using Esri ArcGIS Pro 3.1.0.

## 3. Results

### 3.1. Health Records Involving DBS

A total of 59,111 health records from 48,599 unique persons were identified as involving DBS across the three health datasets between 2011 and 2019 (NHS 24 and A&E attendances) or 2007 and 2019 (SMR01 hospital admissions). This excluded 778 records unrelated to DBS that were removed from the NHS 24 calls and 1 duplicate record that was removed from the SMR01 hospital admissions. There were 22,509 NHS 24 call records from 21,739 persons, 28,096 A&E attendance records from 26,490 persons, and 8506 SMR01 hospital admission records from 7570 persons. Some individuals (7216) appeared in more than one of the datasets.

Much of the research on dog bites has used clinical codes such as ICD-10 codes to identify injuries caused by dogs. The use of these codes means it is not possible to separate dog bites from dog strikes. For this study, it was possible to separate dog bites and dog strikes using the free text information within the NHS 24 call records. This showed that most of the NHS 24 records indicated that they involved only bites (21,535, 97.7%), but some recorded just strikes (548, 2.4%) or both bites and strikes (426, 1.9%).

### 3.2. Yearly Rates

The annual rate of records per 10,000 population in Scotland was calculated (Figure 1). Information for 2019 was excluded, as this represented an incomplete year. The rates of NHS 24 calls per year appear relatively stable, whilst there is a rise for both A&E attendances and SMR01 hospital admissions. Rates of hospital admissions for DBS show a moderate but steady increase over time, whilst for the A&E attendances, the increase in rates appears particularly marked from 2015 onwards. This can largely be attributable to improvements in data recording quality for the specific variable used to extract DBS records for A&E attendances in this study. The variable used—‘Presenting Complaint’—is an optional field, and its completeness (i.e., the percentage of submitted records with non-missing data) has improved significantly over time, rising from just 18% of records in 2011/12 to 95% of records in 2018/19 [41].

Furthermore, information for specific NHS health boards across Scotland (local NHS organisations responsible for planning and delivering healthcare services within their respective regions, with 14 across the country) suggests that the rise in DBS from 2015 onwards is influenced by the inclusion of large areas that did not initially submit this data. For instance, NHS Greater Glasgow & Clyde—Scotland’s largest health board—did not begin submitting data on ‘Presenting Complaint’ until 2016/17, resulting in a substantial undercount of DBS records in earlier years. Additionally, within health boards, not all sites consistently provided data, and some may have submitted incomplete information. For example, within NHS Greater Glasgow & Clyde, data completeness did not exceed 90% until 2017/18 [41].

This suggests that, while A&E attendance for DBS may have genuinely increased, the sharp rise in recent years is largely the result of improvements in data collection and completeness, rather than a sudden surge in DBS injuries. As a result, figures from before 2017/18, when many DBS records from larger health boards were missing, do not provide a reliable estimate of national rates. Consequently, they should not be used to draw conclusions about regional variations or trends over time.

### 3.3. Age

The average (mean) age of individuals with DBS records was the lowest for NHS 24 calls at 30.3 years, with a standard deviation (SD) of 20.7 years, followed by A&E records with a mean age of 36.5 years (SD = 21.2 years), and the highest for SMR01 data with a mean age of 39.3 years (SD = 23.9 years).

Age-specific rates per 10,000 persons were derived for children under one, one to four years, and then at five-year intervals up to 85 years and above (Figure 2).

Rates are the highest for children aged one to nine years, while children under one year typically have lower rates. In all health datasets, the rate of DBS health records is lower around the teenage years of 15–19 years before rising again into adulthood. Rates decline again for older adults in the NHS 24 calls and A&E attendances, but not for the SMR01 hospital admissions, which appear relatively consistent across adult age groups.

### 3.4. Age by Year

The yearly age-specific rates of DBS records for the NHS 24 and SMR01 hospital admissions indicate that the age distribution of DBS records has changed over time (Figure 3). A&E attendances are excluded due to the data quality issues observed over time. Rates are shown by three age groups—14 years and under, 15–69 years, and 70 years and over.

Rates are highest for the youngest age group at the start of the study period (2007 or 2011), but as time increases, the gap between rates for the younger and older age groups appears to be reducing.

For the NHS 24 calls involving DBS, the rate of children aged 14 years and under had fallen between 2011 and 2018, whilst rates had increased for 15–69 year olds and for older adults aged 70 years plus.

SMR01 admission rates for DBS show a rise in adult rates to a much larger degree than the rise seen in rates for children, with all groups converging by the end of 2018.

### 3.5. Age by Incident Location

The age of individuals and the location of the incident were recorded in 12,026 (42.8%) of the 28,096 A&E attendance records, and in 1806 (21.2%) of the 8506 SMR01 hospital admission records. The most common location for incidents to take place was the home or place of residence (Figure 4 and Figure 5). This was most apparent for children in both the A&E and hospital admissions and for older adults in the A&E attendances.

### 3.6. SIMD Decile

The rate of DBS records per 10,000 persons by SIMD decile shows that, for all three types of health data, there is a clear social gradient, with DBS rates falling in line with decreasing levels of deprivation (Figure 6). The rate of NHS 24 calls was 2.5 times higher for individuals living in the most deprived areas compared to the least deprived areas and 2.4 times higher for A&E attendances. The steepest gradient was observed within the SMR01 hospital admissions, where the rate of admissions was 3.9 times higher in the most deprived areas compared to the least deprived areas.

### 3.7. SIMD by Year

The yearly rate of DBS records per 10,000 persons by SIMD quintile for the NHS 24 and SMR01 hospital admissions indicates variation in rates by SIMD quintiles (Figure 7). Quintiles are used over deciles due to low numbers, and A&E attendances are excluded due to data quality issues over time. The difference (ratio) in rates per 10,000 persons between the most and least deprived SIMD quintiles by year is also presented (Figure 8). Hospital admission rates grew most substantially for patients in the most deprived areas (Figure 7b), although the relative gap between SIMD categories does not change in a consistent way over the time period (Figure 8).

### 3.8. SIMD Decile by Age

The age-specific rates of DBS records by SIMD show that the social gradient is consistent for children 14 years and under and in adults up to 70 years, but it is not apparent for adults aged 70 years and above (Figure 9). The global patterns of association between rates of DBS and SIMD may, therefore, be largely driven by younger individuals, particularly children.

The difference (ratio) in rates between the most and least deprived deciles by age group is summarised by dataset (Table 1). For the SMR01 admissions, the gap is the same for children aged 14 years and under and adults under 70 years. However, for the NHS 24 calls and A&E attendances, this gap is the largest for children.

### 3.9. SIMD Domains

The rate of DBS records by SIMD domain-specific deciles shows a clear social gradient across several domains (Figure 10). Similar to the pattern observed using the overall SIMD measure, rates were notably higher for individuals from the most deprived areas, particularly for the income, employment, health, and education domains. For these domains, the rate of records involving individuals from the most deprived areas is at least 2–3 times higher than those from the least deprived areas (Table 2). This difference was most pronounced for the SMR01 admissions.

### 3.10. Local Authority Variation and SIMD

Since health delivery and policy are organised geographically, it is also helpful to explore these patterns at the level of Scotland’s 32 local authorities (LAs). The average annual rate of DBS health records is shown alongside the percentage of datazones within each LA amongst the most and least deprived 20% across Scotland (Figure 11). This measure is known as the ‘local share’, and it can be used to help compare the levels of relative social deprivation between LAs [42]. The average annual rate for A&E attendances by LA was restricted to the years 2017 and 2018 due to known issues in data quality by geographic areas before this point.

Whilst a few LAs feature consistently at the higher or lower end, for many, there is variation across each of the three types of health data. The average annual rate of NHS 24 records was 4.5 (SD = 1.4) per 10,000 persons, ranging from 2.1 in the combined area of the islands (Eilean Siar, Orkney and Shetland) to 7.6 in Midlothian. The average annual rate of records for A&E attendances for DBS was 9.3 (SD = 5.9), ranging much more across LAs, from 0.4 per 10,000 persons in East Ayrshire to 22.7 in West Dunbartonshire. SMR01 admissions due to DBS had an average rate of admissions of 1.2 (SD = 0.4) per 10,000 persons, spanning from an average of 0.5 per 10,000 persons in the combined island LAs to 2.4 in West Lothian.

Some of the areas with higher levels of deprivation also have higher rates of DBS health records. For example, Glasgow City, which has relatively high DBS rates within both the A&E and SMR01 data, also has the largest percentage of areas that are amongst the most deprived. This is similar for West Dunbartonshire and North Ayrshire for the NHS 24 and SMR01 records and Inverclyde for the A&E records. However, this does not apply to all. For example, Midlothian and West Lothian show the highest average rates for NHS 24 and A&E data, respectively, but do not rank highly in area-based deprivation. Likewise, Dundee City is nearer the top end of LAs with the most deprived areas but is nearer the middle in terms of rates of records for all three types of health data.

## 4. Discussion

This study examined patterns in medically attended DBS across Scotland using three health service datasets: NHS 24 calls, A&E attendances, and hospital admissions. Between 2007/2011 and 2018, rates of DBS increased in hospital admissions and A&E attendances, while NHS 24 calls remained relatively stable. The largest increase was seen for the A&E records—although this is largely driven by increases in data quality, rather than a surge in DBS attendances. These findings are consistent with trends of hospital admissions involving DBS observed in England [3], Wales [2], and Ireland [43]. A recent study has also shown this trend in Scottish hospital admissions for DBS for a longer period between 1997 and 2022, showing that, despite a drop in admissions around COVID-19, admissions have continued to rise [1].

### 4.1. Age Patterns

There were clear patterns by age, with DBS rates highest for children aged one to nine years in all health datasets. Young children may struggle to accurately interpret dog body language and stress signals, putting them at greater risk [44]. However, as they get older, children may become more skilled at understanding dog body language and more aware of safe behaviours around pets. Despite this, as Baatz et al. [45] note, even when children recognise signs of fear or stress in dogs, they may not always stop the interaction or behaviour that could lead to a DBS incident, highlighting the importance of vigilant adult supervision.

Over time, rates for children remained relatively stable, while rates for adults increased, narrowing the age gap. This is comparable to other research including hospital admission for DBS across England [3], Wales [2] and Ireland [43]. Possible reasons suggested for this have included changes in children’s outdoor activity [46], changes in family structures or dog ownership patterns over time [3], and the impact of child-focused prevention initiatives [3,47], although evidence of the long-term effectiveness of such interventions is limited [12].

When the location of the incident was recorded, these occurred most frequently at a place of residence or in the home, particularly for children. This is similar to previous research, which has found that most dog bites occur within the home environment or involve a known dog (e.g., [17,19,28,48,49,50,51,52,53,54,55,56,57,58,59,60,61]).

### 4.2. Social Inequalities

There was a clear social gradient in all three health datasets, with rates of health records for DBS at least 2–3 times higher for individuals from the most compared to the least deprived areas. The steepest gradient was observed within the SMR01 hospital admissions. This mirrors patterns seen in England and Wales [3,14,15,16], along with a recent study exploring hospital admissions from DBS in Scotland, which reported a consistent social gradient spanning from 1997 to 2022 and demonstrating the persistence of these inequalities [1].

The strongest gradients were observed for domains closely tied to social disadvantage, such as income, employment, education, and health. Differences between the most and least deprived areas were less for the housing and crime domains, and geographical access to services showed no clear relationship. As an area-based measure, SIMD is unlikely to capture all relevant aspects of inequality, and in Scotland, it is already thought likely to underestimate the experience of deprivation in rural areas [62]. Nevertheless, the patterns with SIMD measures strongly support expectations of a social patterning in DBS incidence.

Inequalities were most apparent for children and adults up to 70 years, with no clear relationship to levels of area-based deprivation for the older adults aged 70 years and above. Children from less advantaged households are often seen as more at risk of experiencing UIs due to a number of risk factors relating to the child and the physical and social environment in which they live and grow up [63]. The interaction between age and social inequality in DBS incidents in Scotland makes a strong case for thinking about social inequalities in DBS prevention strategies.

The regional variation between Scottish local authorities that was evident in each dataset has similar characteristics to patterns reported for England and Wales [2,3]. Geographic variation across local authorities further underscores the complexity of DBS risk. While area-based deprivation may account for some of this variation, other contextual factors—including local enforcement, public awareness, service availability, dog population characteristics, and household circumstances—are also likely contributors.

Social inequalities in DBS incidents may also reflect inequalities in material factors. Studies of dog bites have suggested contributing factors such as limited access to veterinary and behavioural support [2,17,23,48,54,64], disparities in human healthcare access and literacy [19,27,65,66,67], and the acquisition of dogs from lower-welfare sources [2,27]. Environmental and socioeconomic conditions—such as confined living spaces, limited child supervision due to work obligations, and lack of access to safe environments—may also increase risk [2,19,22,64,68,69].

Some research suggests that social patterning in dog ownership—such as higher ownership rates among families with children in more deprived areas—may mediate DBS risk [70,71,72,73], though the findings are mixed [74,75,76]. Others point to the types of dogs owned and their intended use (e.g., protection and status), with breed often implicated despite weak evidence linking breed directly to bite risk [19,22,23,27,64,67,77,78,79].

Behavioural explanations related to inequalities include poorer dog control, lack of training or neutering, and non-compliance with dog control laws [20,48,54]. The supervision of children is often highlighted as key, especially in less advantaged households where factors like single parenthood, caregiver stress, and larger families may reduce supervision capacity [12,19,20,29,45,54,67,80,81,82,83]. Broader structural inequalities are sometimes recognised, highlighting the role of upstream determinants of health [18,26,30,84,85].

### 4.3. Implications

In Scotland and across the UK, there is ongoing interest in ways to reduce the risk of DBS occurring. Significant attention has been given to recent changes in UK-wide legislation with the addition of XL Bully dogs to the list of ‘dangerous dogs’ as part of the DDA (1991, s1), which has been widely contested (e.g., [86,87,88]).

Similar to other parts of the UK, the findings from this research show that medically attended DBS are increasing across Scotland and that there are significant social inequities in their occurrence. However, current policy and prevention efforts across the UK do not explicitly acknowledge these inequalities. This omission is concerning, as policies that fail to directly recognise social inequality risk exacerbating existing inequities [7]. For example, UK-wide policies restricting so-called dangerous breeds may inadvertently worsen inequalities. Inadequate exercise and socialisation opportunities can contribute to increased canine anxiety, which in turn increases the risk of DBS occurring [2]. Restrictions on off-lead exercise in public spaces are likely to disproportionately impact communities with fewer resources to meet a dog’s needs, such as access to private gardens or paid off-lead exercise areas that require booking, incur a cost to use, and may not be easily accessible on foot.

Policy and prevention strategies within the UK are currently centred around ideas of ‘responsible dog ownership’ and the criminalisation of ‘dangerous dogs’ and ‘irresponsible owners’. These policies do not consider the role of wider, systemic factors or ways in which families can be supported to reduce the risk of DBS occurring—especially within the context of rising levels of poverty within the UK.

More supportive, compassion-focused, and family-centred prevention strategies may lead to more effective risk reduction than those attributing blame to the ‘irresponsible’. For example, Pearce [89] suggested that childhood injuries could be reduced by focusing on interventions which improve levels of social support and maternal mental health. Other researchers exploring dog bite risk have also argued that a move away from punitive measures may be more effective in dog bite prevention strategies (e.g., [17,30,32]). Arguably, the most effective policies could involve ‘upstream’ interventions that target the fundamental causes of social disparities [29,48,90,91].

One strategy could be to adopt a public health approach that considers the wider social determinants of health as fundamental drivers that should be tackled via multi-agency collaboration across local government, police, health services, animal welfare organisations, and other relevant stakeholders. At present, although DBS are often claimed to be a public health issue, the underpinnings of this approach are not typically applied in practice [30]. It may also be helpful to consider the intrinsic links between human and canine welfare [34,92,93,94]. A focus on primary intervention through the improvement of human social, economic, and environmental conditions will impact dogs who also share these environments and live within these circumstances. This is an important message for changemakers, whether their interests align most with human or non-human health and welfare.

Education programmes for adults and children are often suggested, yet the effectiveness of some of these interventions has been questioned [12,18], and it has been argued that these strategies are often overused in injury prevention efforts [8]. These methods, if not carefully applied, can also appear as victim-blaming [69], which can add to stigmatisation and discourage help-seeking [95]. Despite these concerns, educational approaches have shown some promise for reducing dog bite risk by teaching children safe behaviours around dogs, recognising risky situations, and accurately interpreting dog body language and stress signals [44,96]. For instance, in a meta-review of child-directed dog safety interventions, Shen et al. [96] found that cognitive/behavioural programmes had the potential to demonstrate moderate improvements in children’s knowledge of dog safety and behaviour around dogs, particularly for interventions that used video stimuli and demonstrations with live animals. In adults, there is also some evidence that similar interventions may be effective for teaching safer behaviours around dogs [97] and recognising stress signals that can lead to a bite occurring [44]. Therefore, well-thought-out and evidence-informed education programmes may be a valuable component of a broader prevention strategy but should not be considered a panacea or the sole measure for reducing risk.

The current UK legislative policy predominantly relies on the identification and reporting of dog control issues and ‘dangerous dogs’ to initiate enforcement actions. However, dog bites and related incidents may be less likely to be reported if they occur within the home or involve a known dog [2,98,99]. This means that the current legislation may be less effective in preventing injuries occurring within this environment [2], which, as shown in this and other research, constitutes the majority of medically attended DBS—particularly for children. To address this, intervention programmes targeting home-based dog bite incidents require further development. Such programmes could benefit from the involvement of agencies with regular family contact, such as health visitors and social workers [2], and be universally implemented in schools [69]. Health care settings outside the home may also provide opportunities to reach families, such as in A&E departments, which were the most frequently attended medical service for DBS within this research.

Research for policy and prevention strategies in DBS that draw on existing models of child injury prevention and UIs research more generally may be particularly helpful, given the extensive body of knowledge that already exists on this topic. There is potential for considerable learning by drawing on the more developed evidence base for reducing inequalities in this area. For example, some researchers within the field of DBS prevention (e.g., [12,26,100,101]) have begun to advocate for similar risk prevention strategies as used in child injury prevention, such as environmental modification strategies and the use of principles such as the ‘Haddon Matrix’ [102], which considers the risk of injury in terms of the host (human), the agent (dog), the physical and social environment, and the stages at which intervention might occur.

Owczarczak-Garstecka et al. [2] highlight that financial barriers to accessing veterinary and behavioural support could contribute to the higher rates of DBS in more deprived areas. This observation is reinforced by Nurse et al. [32], who, through interviews with stakeholders such as police, local authorities, and a small sample of dog owners, identified concerns about the financial cost of addressing ‘dog control’ issues. This is an important point, and it signifies the recognition of the additional challenges faced by dog owners with less financial resources. Broader social and cultural factors may also play a role, and it might be helpful to draw on existing models around barriers in accessing health interventions for lower socioeconomic status groups more widely to inform future research and intervention strategies (e.g., [103,104]).

### 4.4. Limitations and Future Directions

Within the UK, there is a lack of evidence-based policy that considers the extent of DBS or the nature of social inequities. In Scotland, one of the main findings of a review of The Control of Dogs (Scotland) Act 2010 was that a lack of data meant it was not possible to accurately assess the effectiveness of the legislation [5]. This paper, therefore, offers important insights into trends in medically attended DBS in Scotland up to 2018/19. However, A&E data in this study before 2017/18 is incomplete and inconsistent at a national level due to variations in data recording practices and submission rates across health boards. Updated figures also show fluctuations in data quality in more recent years from 2019 onwards [41]. This significantly limits its reliability for analysing trends over time, particularly in earlier years, and any interpretations based on this data must be treated with caution.

The evidence base in this study concerns recorded DBS incidents in health datasets, but it is important to recognise that not all dog bites or strikes will result in an injury, and not all injuries from dogs will result in engagement with health services. A survey in England found that only a third of dog bites resulted in medical attention, with just under 20% presenting to A&E and less than 1% leading to a hospital admission [105]. This highlights an important limitation when using health data, as many dog bite incidents will not be captured through these methodologies.

This limitation is particularly relevant for A&E data. Public Health Scotland (the national-level ‘health board’) will not report on DBS statistics from A&E using ICD-10 codes alone because they believe the data are not robust enough [106]. Clinical coding within the A&E records is carried out by already overburdened and/or inadequately trained clinicians, while in hospital admissions records, this is carried out by specialist staff, resulting in higher-quality data [107]. To address this, we enhanced case identification by supplementing ICD-10 codes with free text searches, resulting in a much higher capture rate of DBS attendances to A&E and preparatory work in this study showed that around 80% of the A&E attendances identified as DBS would have been missed had ICD-10 codes been the only method used to identify these. However, the free text field was optional and inconsistently completed across years and regions, meaning that some cases were still likely missed. Therefore, while our approach achieves an improvement over standard methods, limitations in the completeness and consistency of A&E data remain and must be considered when interpreting these findings.

Another limitation of using health records is the lack of contextual information surrounding the DBS incident. Within the NHS 24 data, a free text variable contains small amounts of contextual information. However, this was limited, inconsistent, and unsuitable for use in any robust way for analytical purposes. More research, for instance, through the co-production of studies that actively involve communities most at risk, is desirable to explore contexts further. Engaging these communities directly will help ensure their voices are heard and interventions tailored to their specific needs and challenges.

Our empirical conclusions are based solely on cases recorded within the included health datasets, and their applicability to all dog bite incidents rests on two key assumptions: first, that the factors that determine whether or not medical assistance is sought after a given incident are unrelated to the phenomena that we are analysing, and second, that the factors determining whether an incident is accurately recorded as dog-related within administrative records (including clinical coding and data submission practices) are likewise unrelated to the associations that we explore. The second assumption could be compelling since we anticipate issues such as bureaucratic arrangements are the main driver of what is documented within records, and these should reflect the wider organisation, rather than specific conditions related to individual cases. The first assumption could be harder to assess, but it could be plausible if the severity of injury is the most consequential influence on whether medical help is sought.

In summary, this study has several key unavoidable limitations that should be borne in mind. First, although we examined a large sample of health records, not all DBS incidents result in medical attention, meaning the true burden is likely underestimated. Second, the quality and completeness of A&E data prior to 2017/18 were inconsistent across time and locality, limiting their reliability for assessing regional or temporal trends. Third, contextual information about DBS incidents is largely absent, particularly around the dog involved, and the circumstances of the event. Fourth, reliance on administrative health data introduces selection bias since individuals who seek medical help may differ systematically from those who do not. Fifth, while quantitative data highlight important structural patterns, they cannot capture the depth of understanding that qualitative or participatory research can provide, especially around the lived experiences of families and communities affected by DBS. Finally, this study focuses on Scotland and may not be generalisable to other regions with different social or policy contexts or to more recent periods influenced by changes in lifestyle and dog ownership, particularly following the COVID-19 pandemic. These limitations should be carefully considered when interpreting our findings.

Future directions that may help mitigate some of these limitations could include the use of a wider range of methodologies such as mixed methods designs that can help to capture valuable qualitative insights from victims, families, professionals, and communities most at risk. The evaluation of interventions should be assessed with respect to differentials in the effects of these for different groups [8], and more research is needed to establish specific strategies that are most helpful for reducing dog bite risk in more disadvantaged communities. The voices of those most at risk must be heard to ensure prevention efforts are most effective and do not unintentionally disempower communities [69,91,108]. Collaborations between communities and external changemakers may also help build trust [17], and the inclusion of community members through the co-production of research will help changemakers better understand the pathways leading to inequalities in DBS and the barriers preventing change.

More creative and innovative solutions may also be useful to help gain richer insights, such as emerging technologies using virtual reality or simulated programmes that can help serve as educational tools and research methodologies to assess and/or modify individual responses to situations that might otherwise involve safety concerns for participants and animal welfare issues [109]. Examples of this include educational games such as The Blue Dog [110], the virtual reality tool DAVE (Dog Assisted Virtual Environment) [111], and video simulators [97] displaying a range of dog behaviours. Additionally, the use of mobile phone apps and online platforms could assist in data collection and education efforts, which may be more widely and easily accessible for certain groups.

Further research may also wish to consider maximising the use of other sources of existing data such as the newly created Scottish DCN Database for LAs and police across Scotland [112], which could provide additional opportunities to explore information related to DBS incidents from records of DCNs and incidents reported to authorities. In addition, despite the limitations surrounding administrative health data, hospital admission statistics could be used to track inequalities internally (through the NHS Scotland infrastructure) and externally if made more accessible in the public domain. This could be facilitated by separating hospital admission figures for DBS from ‘other’ types of unintentional injuries (UIs) in annual PHS statistical reports and by providing breakdowns by SIMD.

Based on the findings of this paper, the following actionable recommendations are proposed:The acknowledgement of social inequities in DBS within policy and prevention strategies and in research priorities, alongside the review of all new and existing polices to ensure that they do not perpetuate these inequities.The improvement of data surveillance so that inequities can be monitored and policies can be evidence-informed and evaluated to ensure that they are effective both in reducing DBS risk and narrowing social inequality. This might include the following:
oInvesting in improvements for recording DBS within health records, particularly for A&E records which have known data quality issues. This might be achieved by providing additional resources and training and by ensuring that efficient systems are in place to assist already overburdened health care staff (e.g., by using specialist clinical coders for A&E data).oMaximising the use of existing statistics. For instance, PHS may wish to consider separating DBS figures from ‘other’ types of UIs in their statistical publications to allow for improved data surveillance. Other options include exploring the use of the newly created Scottish DCN Database for research and data monitoring purposes. It may also be helpful to explore opportunities for data linkage between existing data sources where possible.
Ensure that prevention strategies are compassion-focused and family-centred. Policies to reduce risk should be supportive, rather than focusing on criminalisation or attributing blame to ‘irresponsible’ owners. Policies should be designed to assist, rather than penalise.Utilising the potential of a public health approach that includes the consideration of the wider social determinants of health and a multi-agency approach exploring systemic factors and focusing more on upstream intervention.Recognise the issue of DBS as a One Health concern where the interlinked nature of pet health and welfare with human health and welfare necessitates their joint consideration in broader health and social policies.A greater focus on DBS as a type of UI. This could involve drawing insights from the more developed literature on health inequalities in other types of UIs and exploring how these findings can inform the understanding and prevention of DBIs.Make room for the voices of the populations most at risk to better understand the pathways which lead to these inequalities and the barriers preventing change. Encourage positive change through the co-production of research and prevention strategies to empower communities.

## 5. Conclusions

The results reported above use health data across Scotland for a period covering nearly 13 years to explore population-level trends in medically attended DBS and identify significant social inequities in their occurrence. The use of three health data sources also provided novel insights into the extent of DBS at different levels of health service engagement.

In this study, dog bites and strikes were conceptualised as a type of preventable unintended injury (UI) and expected to show similar social inequalities, as have been reported for other UIs in Scotland [63]. Viewed as a type of UI helps us recognise social inequities in DBS. Policies to reduce DBS might gain from a reduced focus on dog breed legislation and punitive measures and more attention to social and environmental factors at the individual and systemic level.

## 6. Patents

This work contains statistical data from the National Records of Scotland (NRS), which is Crown Copyright. The use of NRS statistical data in this work does not imply the endorsement of the NRS in relation to the interpretation or analysis of the statistical data. This work uses research datasets which may not exactly reproduce National Statistics aggregates.

## Figures and Tables

**Figure 1 animals-15-01971-f001:**
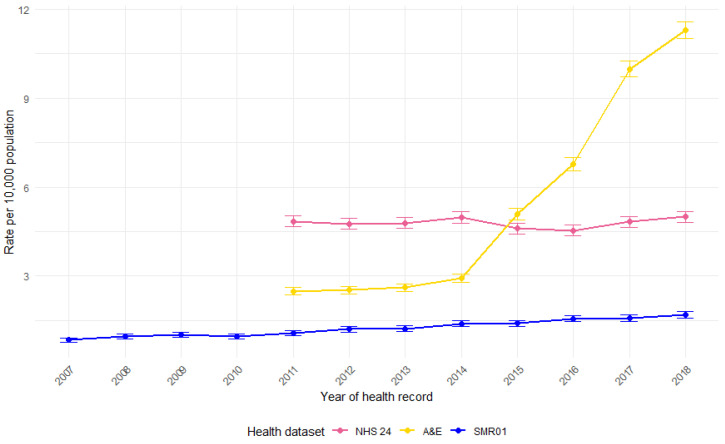
Rate of health records involving DBS per 10,000 population in Scotland, by year and health dataset (NHS 24 calls, A&E attendances, and SMR01 hospital admissions), with 95% confidence intervals.

**Figure 2 animals-15-01971-f002:**
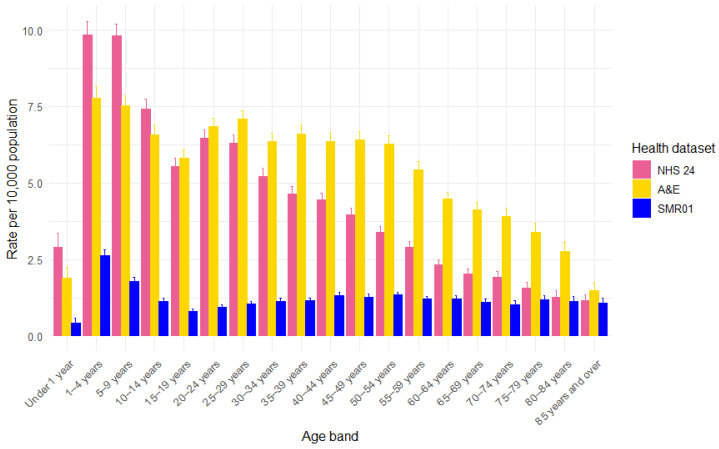
Rate of health records involving DBS per 10,000 population in Scotland, by age band and health dataset (NHS 24 calls, A&E attendances, and SMR01 hospital admissions), with 95% confidence intervals.

**Figure 3 animals-15-01971-f003:**
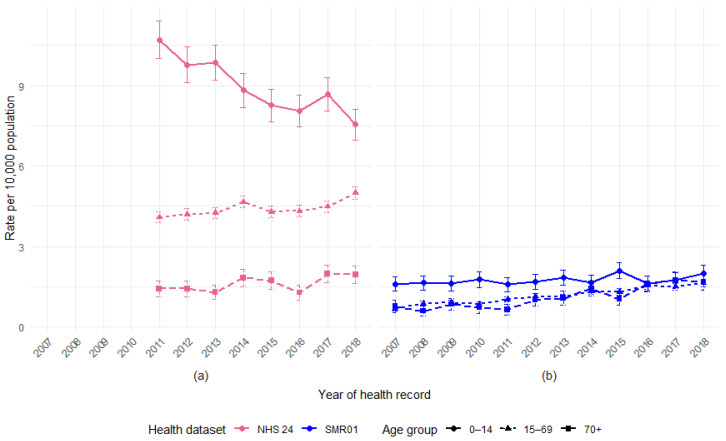
Rate of health records involving DBS per 10,000 population in Scotland, by year and age group, for (**a**) NHS 24 calls and (**b**) SMR01 hospital admissions, with 95% confidence intervals.

**Figure 4 animals-15-01971-f004:**
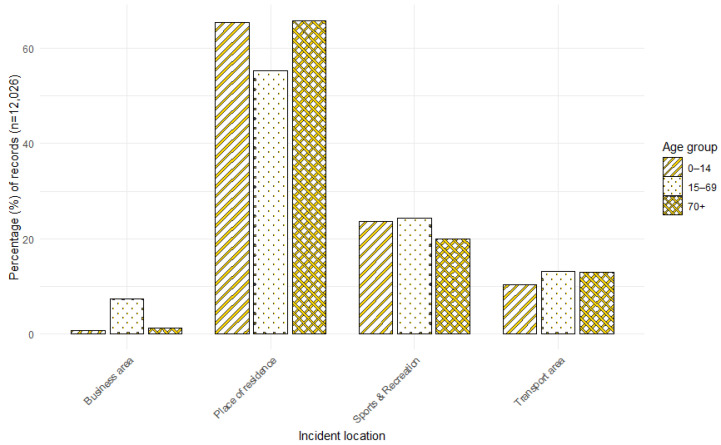
Percentage distribution of A&E attendance records for DBS by incident location and age.

**Figure 5 animals-15-01971-f005:**
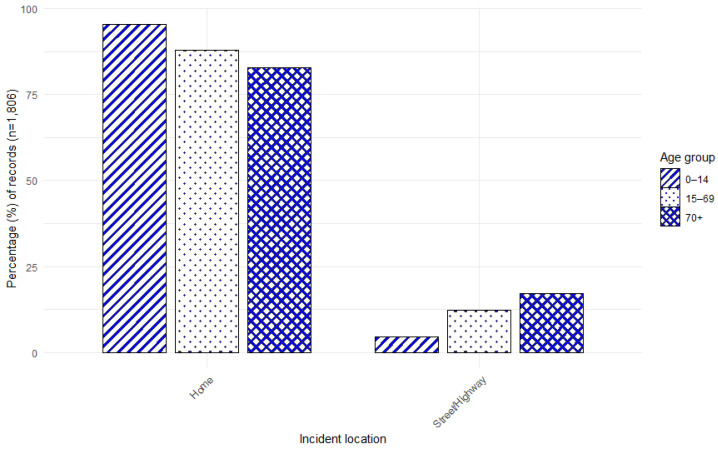
Percentage distribution of SMR01 hospital admission records for DBS by incident location and age.

**Figure 6 animals-15-01971-f006:**
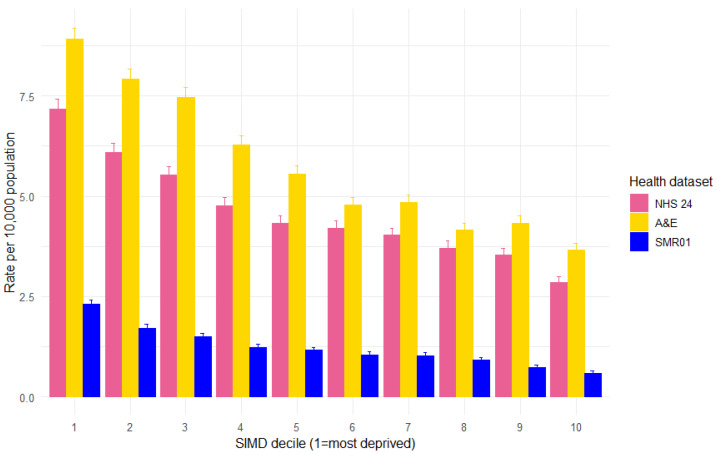
Rate of health records involving DBS per 10,000 population in Scotland, by SIMD decile and health dataset (NHS 24 calls, A&E attendances, and SMR01 hospital admissions), with 95% confidence intervals.

**Figure 7 animals-15-01971-f007:**
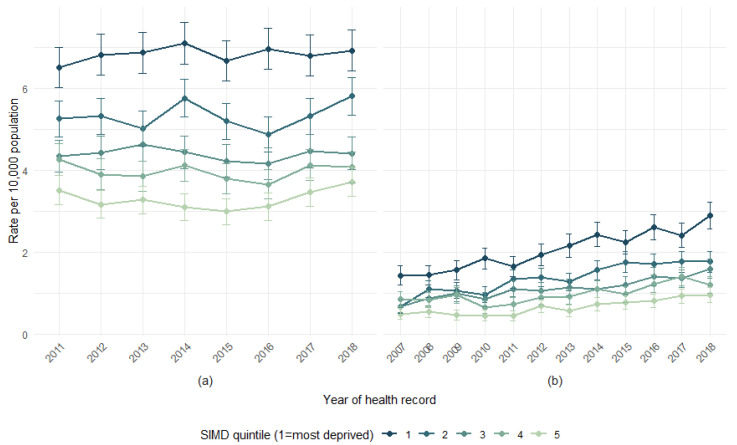
Rate of health records involving DBS per 10,000 population in Scotland, by year and SIMD quintile, for (**a**) NHS 24 calls and (**b**) SMR01 hospital admissions, with 95% confidence intervals.

**Figure 8 animals-15-01971-f008:**
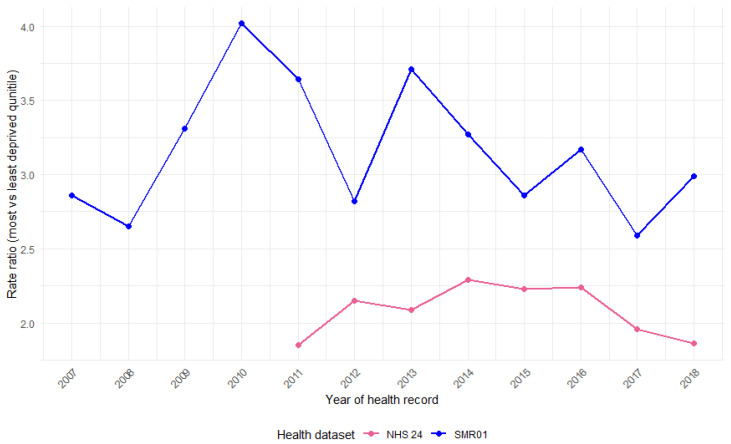
Difference (ratio) in rates per 10,000 population in Scotland between the most and least deprived SIMD quintiles, by year and health dataset (NHS 24 calls, A&E attendances, and SMR01 hospital admissions).

**Figure 9 animals-15-01971-f009:**
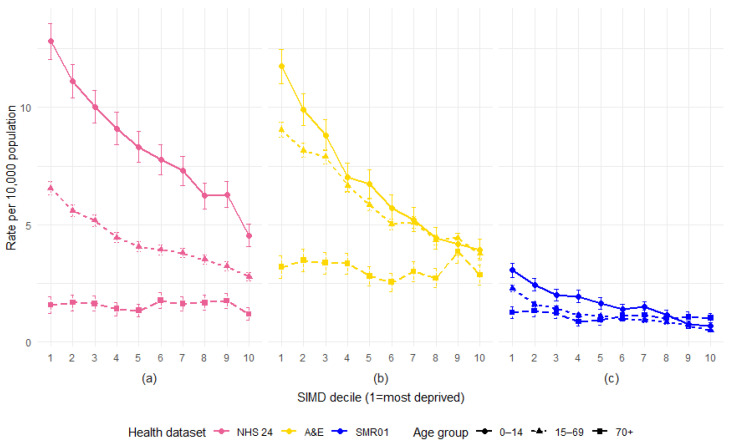
Rate of health records involving DBS per 10,000 population in Scotland, by SIMD decile and age group, for the following: (**a**) NHS 24 calls; (**b**) A&E attendances; and (**c**) SMR01 hospital admissions, with 95% confidence intervals.

**Figure 10 animals-15-01971-f010:**
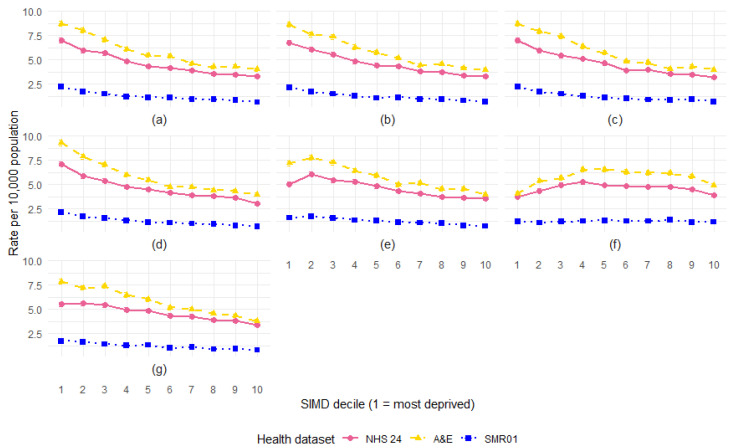
Rate of health records involving DBS per 10,000 population in Scotland, by SIMD decile for each domain: (**a**) income; (**b**) employment; (**c**) health; (**d**) education; (**e**) housing; (**f**) access; and (**g**) crime.

**Figure 11 animals-15-01971-f011:**
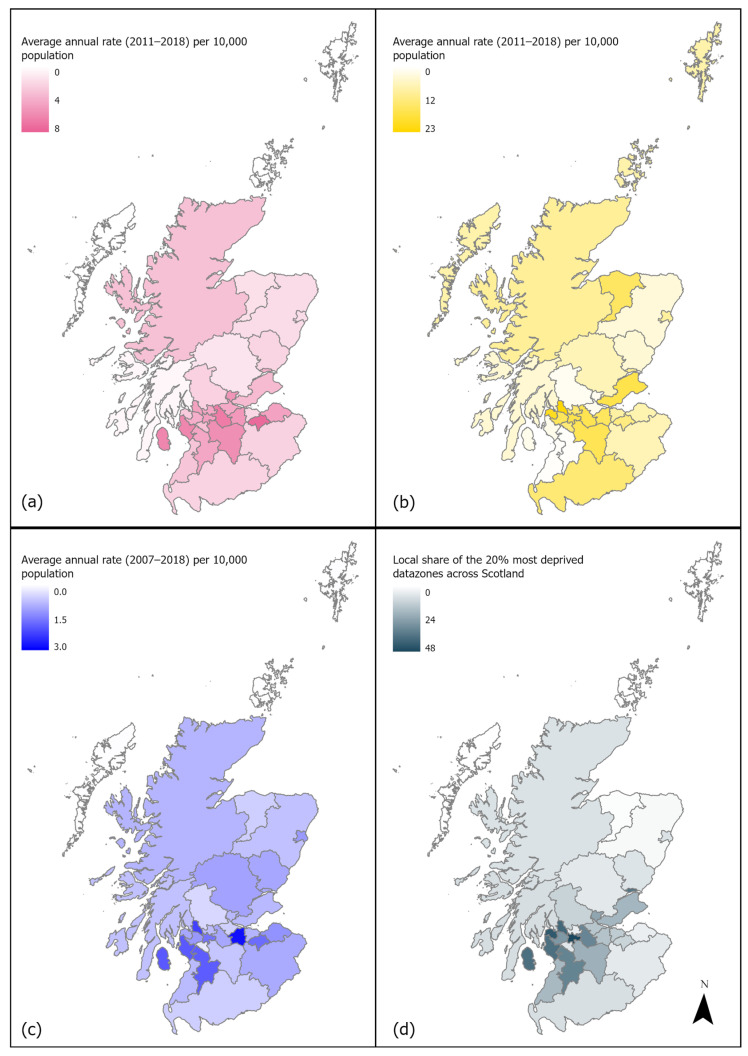
Local authority average annual rates of health records involving DBS per 10,000 population in Scotland for the following: (**a**) NHS 24 calls; (**b**) A&E attendances; (**c**) SMR01 hospital admissions; and (**d**) local authority local share of the 20% most deprived datazones across Scotland. Created in Esri ArcGIS Pro 3.1.0 using local authority boundary files [40] licensed under the Open Government Licence v.3.0; Contains Ordnance Survey data. © Crown copyright and database right 2019.

**Table 1 animals-15-01971-t001:** Difference (ratio) in rates per 10,000 population in Scotland between the most and least deprived SIMD quintiles, by age group and health dataset (NHS 24 calls, A&E attendances, and SMR01 hospital admissions).

Age Group	NHS 24 Calls	A&E Attendances	SMR01 Admissions
0–14	2.8	3.0	4.6
15–69	2.4	2.4	4.6
70+	1.3	1.1	1.2
All ages	2.5	2.4	3.9

**Table 2 animals-15-01971-t002:** Difference (ratio) in rates per 10,000 population in Scotland between the most and least deprived SIMD quintiles, by SIMD domain and health dataset (NHS 24 calls, A&E attendances, and SMR01 hospital admissions).

SIMD Domain	NHS 24 Calls	A&E Attendances	SMR01 Admissions
Income	2.1	2.2	3.4
Employment	2.0	2.2	3.0
Health	2.2	2.2	3.2
Education	2.3	2.3	3.1
Housing	1.4	1.8	2.2
Access	1.0	0.8	1.0
Crime	1.6	2.1	2.1

## Data Availability

Data is unavailable due to privacy and ethical restrictions—please see the methods section.

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
