# Peer review of "Social Inequalities in Dog Bites and Strikes in Scotland: Evidence from Administrative Health Records and Implications for Prevention Policy"

_animals, 2025, doi:10.3390/ani15131971_

Round 1
Reviewer 1 Report
Comments and Suggestions for Authors
Comments on the manuscript “Social Inequalities in Dog Bites and Strikes in Scotland: Evidence from Administrative Health Records and Implications for Prevention Policy”, submitted to Animals
This manuscript analyzes the incidence of dog bites and strikes in Scotland between 2007-2019. Based on the analysis of three databases, the incidence of dog bites and strikes was higher in deprived areas. The causes of these differences are many and are not easily reduced without social, behavioral, economic, legal and political interventions. The manuscript suggests some of these interventions.
Public health neglects dog bite accidents and the social and economic context of deprived areas. Therefore, the study's topic is relevant, demonstrated in a well-written text, with rigorous methods and a discussion with broad coverage of the results.
The text generally allows a broad understanding of the observations and analyses. The methods are rigorous and relevant to the study's objective. The results are well presented, although I doubt the need for Figure 8.
The discussion is long, but it is necessary due to the complexity of the analyses and results. References are necessary and support the arguments.
The manuscript has no spelling errors, except for three minor errors. Line 37 is missing the keywords. Lines 111 to 121 (2.1 Defining dog bites and strikes) are a paragraph that should be in the introduction. The sentence between lines 510 and 512 is confusing.
Author Response
Comment 1: Line 37 is missing the keywords.
Response: Thank you for pointing this out. Keywords have now been added between lines 37 and 39 in the revised manuscript.
Comment 2: Lines 111 to 121 (2.1 Defining dog bites and strikes) are a paragraph that should be in the introduction.
Response: Thank you for the suggestion. This paragraph has been moved to the introduction (now lines 94–105), and subsection numbers in Section 2 have been updated accordingly.
Comment 3: The sentence between lines 510 and 512 is confusing.
Response: Thank you for highlighting this. The sentence has been revised for clarity and now appears on lines 547–549 (or lines 478–480 in the version without tracked changes).
Reviewer 2 Report
Comments and Suggestions for Authors
The researchers investigate the rising incidence of dog bites and strikes (DBS) in Scotland from 2007 to 2019, highlighting social inequalities linked to these incidents. Using administrative health records, they reveal that younger individuals and those in deprived areas are disproportionately affected. While improved data reporting accounts for a notable rise in A&E attendances and hospital admissions due to dog bites, the actual increase in incidents is less certain. Those from deprived areas face a two- to four-fold higher risk compared to affluent individuals.
Most DBS incidents occur at home and involve familiar dogs, stressing the need for targeted prevention strategies. The study indicates a strong correlation between socioeconomic factors—such as income, employment, and education—and the likelihood of experiencing dog bites. The authors argue that current UK policies focusing on breed-specific legislation and punitive measures may not effectively address the root causes of these injuries. They advocate for public health-oriented strategies that engage communities and provide support, along with educational programs to promote safe dog interactions.
The authors acknowledge certain limitations in data completeness and quality, particularly in A&E records prior to 2017. However, several other limitations need to be emphasized by the authors in a consolidated paragraph. Some of these are mentioned, but they must be brought together so that the reader clearly understands them when interpreting the conclusions. The authors also mention that Public Health Scotland does not report on A&E statistics for DBS because the data are not considered robust enough, which raises concerns about the reliability of the findings from this source. Furthermore, the study relies on health records, which may not capture all dog bite incidents, as many do not result in medical attention. Previous research indicates that only a small fraction of dog bites result in hospital visits. There is a lack of contextual information surrounding the DBS incidents recorded in the health datasets. The available free-text variables are limited and inconsistent, making it challenging to comprehensively analyze the circumstances of each incident. The study primarily examines recorded DBS incidents in health datasets, which may not accurately reflect the broader picture of dog bites, as many injuries may go unreported or untreated. The reliance on health records may introduce selection bias, as individuals who seek medical attention for dog bites may differ significantly from those who do not, potentially skewing the results toward more severe cases. The study primarily uses quantitative data from health records, which may overlook qualitative insights that could provide a deeper understanding of the circumstances surrounding dog bites, such as personal experiences, environmental factors, and community attitudes toward dogs. The findings may not be applicable to other regions or populations outside Scotland, as the study is specific to the Scottish context and its unique health and social dynamics. These limitations suggest that while the study provides valuable insights into the patterns of dog bites in Scotland, caution should be exercised when interpreting the results. The study covers data up to 2019, which may not account for more recent trends or changes in policies and societal behaviors regarding dog ownership and interactions, especially in the context of the COVID-19 pandemic.
To guide future prevention efforts, the authors should also include a section on effective educational programs, detailing what has been proven to be effective in reducing dog bite incidents. They can also consider discussing how technological advances (e.g., apps for reporting dog bites or educational tools) can be leveraged to improve data collection and public education. Also, while the paper discusses the need for policy changes, it could benefit from outlining specific, actionable recommendations for policymakers, including examples of successful interventions from other regions or countries.
Finally, the authors should strive to provide clear recommendations for future research, identify specific knowledge gaps, and suggest methodologies for addressing them.
The data at their disposal should be enough to address the points I have suggested.
Keywords are missing. Please complete.
Please adopt the convention of referencing figures and tables at the end of the relevant sentence. For example, refrain from beginning sentences with “Figure 3 shows the yearly age-specific rates…” Instead, state the content first and then reference it as (Fig. 3). Similarly, delete sentences such as “These are shown in Figure 2.” and simply add (Fig. 2) to the end of the preceding sentence.
Author Response
Thank you for taking the time to review this manuscript and for your helpful comments. Please find the detailed responses below and the corresponding revisions in tracked changes in the re-submitted files. Line numbers are provided with and without tracked changes shown.
Comment 1: The authors acknowledge certain limitations in data completeness and quality, particularly in A&E records prior to 2017. However, several other limitations need to be emphasized by the authors in a consolidated paragraph. Some of these are mentioned, but they must be brought together so that the reader clearly understands them when interpreting the conclusions. The authors also mention that Public Health Scotland does not report on A&E statistics for DBS because the data are not considered robust enough, which raises concerns about the reliability of the findings from this source. Furthermore, the study relies on health records, which may not capture all dog bite incidents, as many do not result in medical attention. Previous research indicates that only a small fraction of dog bites result in hospital visits. There is a lack of contextual information surrounding the DBS incidents recorded in the health datasets. The available free-text variables are limited and inconsistent, making it challenging to comprehensively analyze the circumstances of each incident. The study primarily examines recorded DBS incidents in health datasets, which may not accurately reflect the broader picture of dog bites, as many injuries may go unreported or untreated. The reliance on health records may introduce selection bias, as individuals who seek medical attention for dog bites may differ significantly from those who do not, potentially skewing the results toward more severe cases. The study primarily uses quantitative data from health records, which may overlook qualitative insights that could provide a deeper understanding of the circumstances surrounding dog bites, such as personal experiences, environmental factors, and community attitudes toward dogs. The findings may not be applicable to other regions or populations outside Scotland, as the study is specific to the Scottish context and its unique health and social dynamics. These limitations suggest that while the study provides valuable insights into the patterns of dog bites in Scotland, caution should be exercised when interpreting the results. The study covers data up to 2019, which may not account for more recent trends or changes in policies and societal behaviors regarding dog ownership and interactions, especially in the context of the COVID-19 pandemic.
Response: Thank you for your suggestion to emphasise further some of the limitations of the study in a consolidated paragraph. This has been added to the revised manuscript (lines 691-706 or 645-660 without tracked changes). An additional paragraph has also been inserted immediately before this (lines 679–690 or 633–644) to further emphasise key concerns. We have also expanded the paragraph beginning on line 649 (or 579 without tracked changes) to clarify that the statistics PHS won’t report on are those that include only ICD-10 codes to identify DBS. However, we also acknowledge that even with additional free-text fields, caution is still required when interpreting findings from A&E data.
Comment 2: To guide future prevention efforts, the authors should also include a section on effective educational programs, detailing what has been proven to be effective in reducing dog bite incidents. They can also consider discussing how technological advances (e.g., apps for reporting dog bites or educational tools) can be leveraged to improve data collection and public education.
Response: Thank you for the helpful suggestion. While the manuscript acknowledges mixed findings on the effectiveness of educational programmes, some interventions have shown promise. We have added text and citations reflecting this on lines 589–595 (or 519–525 without tracked changes). We have also included discussion of potential technological advances to enhance both data collection and public education on lines 719–727 (or 641–649 without tracked changes).
Comment 3: While the paper discusses the need for policy changes, it could benefit from outlining specific, actionable recommendations for policymakers, including examples of successful interventions from other regions or countries.
Response: Thank you for this suggestion. We have added several specific, actionable recommendations between lines 737 and 775 (or 659-697 without traced changes).
Comment 4: The authors should strive to provide clear recommendations for future research, identify specific knowledge gaps, and suggest methodologies for addressing them.
Response: Thank you for suggesting we add further recommendations for future research. Additional text has been added to address this between lines 707 and 736 (or 629-658 without tracked changes)
Comment 5: Keywords are missing. Please complete.
Response: Thank you for pointing out the missing keywords. These have been added to the revised manuscript between lines 37 and 39.
Comment 6: Please adopt the convention of referencing figures and tables at the end of the relevant sentence. For example, refrain from beginning sentences with “Figure 3 shows the yearly age-specific rates…” Instead, state the content first and then reference it as (Fig. 3). Similarly, delete sentences such as “These are shown in Figure 2.” and simply add (Fig. 2) to the end of the preceding sentence.
Response: Thank you for this advice. We have revised all references to figures and tables throughout the manuscript to follow this convention, placing them at the end of the relevant sentence (e.g., “...as shown in Fig. 2”).